# Phase Angle and Nutritional Status: The Impact on Survival and Health-Related Quality of Life in Locally Advanced Uterine Cervical Cancer

**DOI:** 10.3390/healthcare11020246

**Published:** 2023-01-13

**Authors:** Faviola González-Barba, Luz-Ma.-Adriana Balderas-Peña, Benjamín Trujillo-Hernández, Luz-María Cervantes-González, Javier-Andrés González-Rodríguez, Leonardo-Xicotencatl Gutiérrez-Rodríguez, Adriana Alvarado-Zermeño, Aldo-Antonio Alcaraz-Wong, Eduardo Gómez-Sánchez, Gabriela-Guadalupe Carrillo-Núñez, Mario Salazar-Páramo, Arnulfo-Hernán Nava-Zavala, Benjamín Rubio-Jurado, Mario-Alberto Mireles-Ramírez, Brenda-Eugenia Martínez-Herrera, Daniel Sat-Muñoz

**Affiliations:** 1Departamento Clínico de Anatomía Patológica, División de Diagnóstico, UMAE, Hospital de Especialidades, Centro Médico Nacional de Occidente, Instituto Mexicano del Seguro Social, Guadalajara 44340, Mexico; 2Unidad de Investigación Biomédica 02, UMAE Hospital de Especialidades (HE), Centro Médico Nacional de Occidente (CMNO), Instituto Mexicano del Seguro Social (IMSS), Guadalajara 44340, Mexico; 3Departamento de Morfología, Centro Universitario de Ciencias de la Salud (CUCS), Universidad de Guadalajara (UdG), Guadalajara 44340, Mexico; 4Cuerpo Académico UDG CA-874 “Ciencias Morfológicas en el Diagnóstico y Tratamiento de la Enfermedad”, Guadalajara 44340, Mexico; 5Doctorado en Ciencias Médicas, Facultad de Medicina, Universidad de Colima, Colima 28040, Mexico; 6Carrera de Médico Cirujano y Partero, Coordinación de Servicio Social, Centro Universitario de Ciencias de la Salud (CUCS), Universidad de Guadalajara (UdG), Guadalajara 44340, Mexico; 7Comisión Interinstitucional de Formación de Recursos Humanos en Salud, Programa Nacional de Servicio Social en Investigación 2021, Demarcación Territorial Miguel Hidalgo 11410, Mexico; 8Carrera de Médico Cirujano y Partero, Coordinación de Servicio Social, Centro Universitario del Sur, Universidad de Guadalajara (UdG), Ciudad Guzmán 49000, Mexico; 9Departamento Clínico de Oncología Radioterapia, Servicio Nacional de Radioneurocirugía, División de Oncología Hematología, UMAE, Hospital de Especialidades, Centro Médico Nacional de Occidente, Instituto Mexicano del Seguro Social, Guadalajara 44340, Mexico; 10División de Disciplinas Clínicas, Centro Universitario de Ciencias de la Salud (CUCS), Universidad de Guadalajara (UdG), Guadalajara 44340, Mexico; 11Departamento de Microbiología y Patología, Centro Universitario de Ciencias de la Salud (CUCS), Universidad de Guadalajara (UdG), Guadalajara 44340, Mexico; 12Cuerpo Académico UDG CA-365 “Educación y Salud”, Guadalajara 44340, Mexico; 13Academia de Inmunología, Departamento de Fisiología, Centro Universitario de Ciencias de la Salud (CUCS), Universidad de Guadalajara (UdG), Guadalajara 44340, Mexico; 14Unidad de Investigación Social Epidemiológica y en Servicios de Salud, Órgano de Operación Administrativa Desconcentrada, Guadalajara 44340, Mexico; 15Programa Internacional Facultad de Medicina, Universidad Autónoma de Guadalajara, Zapopan 45129, Mexico; 16Servicio de Inmunología y Reumatología, División de Medicina Interna, Hospital General de Occidente, Secretaria de Salud Jalisco, Zapopan 45170, Mexico; 17Departamento Clínico de Hematología, División de Oncología Hematología, UMAE, Hospital de Especialidades, Centro Médico Nacional de Occidente, Instituto Mexicano del Seguro Social, Guadalajara 44340, Mexico; 18División de Investigación en Salud, UMAE, Hospital de Especialidades, Centro Médico Nacional de Occidente, Instituto Mexicano del Seguro Social, Guadalajara 44340, Mexico; 19Hospital General de Zona (HGZ) #02 c/MF “Dr. Francisco Padrón Puyou”, Órgano de Operación Administrativa Desconcentrada San Luis Potosi, IMSS, San Luis Potosi 78250, Mexico; 20Comité de Cabeza y Cuello, UMAE, Hospital de Especialidades, Centro Médico Nacional de Occidente, Instituto Mexicano del Seguro Social, Guadalajara 44340, Mexico; 21Departamento Clínico de Oncología Quirúrgica, División de Oncología Hematología, UMAE, Hospital de Especialidades, Centro Médico Nacional de Occidente, Instituto Mexicano del Seguro Social, Guadalajara 44340, Mexico

**Keywords:** phase angle, cervix–uteri cancer, nutritional state, quality of life

## Abstract

The phase angle, an indicator of muscle mass status and membrane cell integrity, has been associated with low survival, poorer clinical outcomes, and worse quality of life among cancer patients, but information on women with uterine cervical cancer (UCCa) is scarce. In this prospective study, we used a bioelectrical impedance analyzer to obtain the PA of 65 women with UCCa. We compared the health-related quality of life and inflammatory and nutritional indicators between low PA and normal PA. The mean age was 52 ± 13. The low PA and normal PA groups differed in terms of the C-reactive protein (15.8 ± 19.6 versus 6.82 ± 5.02, *p* = 0.022), glucose (125.39 ± 88.19 versus 88.78 ± 23.08, *p* = 0.021), albumin (3.9 ± 0.39 versus 4.37 ± 0.30, *p* = 0.000), EORTC QLQ-C30 loss of appetite symptom scale score (33.33 (0.0–100.00) versus 0.0 (0.0–0.0), *p* = 0.005), and EORTC QLQ-CX24 menopausal symptoms scale score (0.0 (0.0–33.33) versus 0.0 (0.0–100.0), *p* = 0.03). The main finding of the present study is the interaction between PA and obesity as critical cofactors in the UCCa adeno and adenosquamous histologic variants, to a greater extent than cervical squamous cell carcinoma.

## 1. Introduction

Uterine cervical cancer (UCCa) is the second most common cancer in women aged over 65 worldwide. However, in emerging economies, such as Latin America, Asia, and Africa, the women affected are younger, and the impact on these women and their families is more profound [1].

A considerable percentage of newly diagnosed cases in developing countries are detected at a locally advanced clinical stage (IB2 to III, according to FIGO staging) or even a metastatic stage. The clinical conditions described above prevent these women from being candidates for standard surgical treatment according to the NCCN and ESGO criteria in the US and EU. They should be treated by concurrent chemoradiotherapy [1,2].

Limited access to health services and high migration rates are common among women with cervical cancer in developing countries. They have limited treatment options, and this condition profoundly affects survival. In some cases, they undergo hysterectomy, associated with radiosensitizing chemotherapy, radiotherapy, and brachytherapy [3,4].

In clinical settings, the health team assumes that malnutrition in UCCa is a common condition that directly impacts body weight [5]. In the case of UCCa, only a few reports identified in our literature search described phase angle (PA) alterations in women with uterine cancer. Nevertheless, these reports highlighted the role of the phase angle as an anthropometric marker for malnutrition and sarcopenia. The phase angle evaluates the loss of muscle mass and the integrity of the membrane cells. Simultaneously, the reports also emphasized the participation of fat mass as a source of inflammatory mediators and molecules involved in malignant transformation and processes such as migration, proliferation, and metastases [6,7,8,9,10,11].

The gold standard for body composition assessment in clinical settings is magnetic resonance imaging (MRI) or computed tomography (CT). However, one of the most efficient methods is bioelectrical impedance analysis (BIA), which is considered an excellent marker of the function of the cellular membrane, with the additional advantage of inferring its integrity, which is considered a global marker of health [12,13]. All of these techniques allow us to identify a loss of skeletal muscle mass, even in patients with overweight and obesity, associated with the phenomenon of sarcopenic obesity.

Even with the best treatment options, clinical outcomes and patient-reported outcomes can be affected by the patient’s nutritional status, including survival rate, functionality, and health-related quality of life (HRQoL), while the causes remain unclear [1,2,13]. However, we do not yet know how profound the impacts of sarcopenic obesity and obesity on UCCa are. There are few literature reports about sarcopenia, sarcopenic obesity, or obesity and their relationships with UCCa adenocarcinoma, in contrast to squamous histologies.

The current report aims to describe the relationship between the nutritional state, phase angle, biochemical markers, functionality, and health-related quality of life in women with UCCa and its histologic type.

## 2. Materials and Methods

The Institutional Review Board of the Instituto Mexicano del Seguro Social (Comisión Nacional de Investigación Científica) approved this study (registration number R-2017-785-038). All procedures were carried out in accordance with the principles of the Declaration of Helsinki, and all the patients signed an informed consent form to participate.

We included a prospective cohort of 65 patients with a histological confirmation of locally advanced UCCa, who were treated with radiotherapy because they were not candidates for surgical treatment, and they were followed for three years. All participants were treated at a tertiary hospital in Guadalajara, Mexico. We excluded patients with (1) two or more malignant neoplasms, (2) AIDS/HIV infection, (3) autoimmune disease, (4) any chronic disease, (5) failure to sign the informed consent form, or (6) any contraindication for performing bioelectrical impedance analysis (weight >300 kg, metal prostheses, electronic implants, edema, or limb amputations).

The patients’ sociodemographic data, information on the clinical stage and anatomical location of the tumor, and information on the treatment modality were obtained from their medical records.

An experienced dietitian performed the body composition analysis. Measurements of height (m) and weight (kg) were taken with a SECA 213 device (Seca, Hamburg, Germany). A BIA device mBCA SECA 514 (Seca, Hamburg, Germany) was used to measure the phase angle (PA), body mass index (BMI), lean mass percentage, and skeletal muscle mass (SMM).

PA is a BIA-subrogated measure that uses the resistance and reactance generated by body fluids and cell membranes (capacitance) in the human body. PA is understood as the opposition of a structure/tissue to the electrical current through the body and the resistance of the cell membrane. The normal phase angle cut-off was set to 4.45°.

The skeletal muscle mass index (SMMI) was calculated using the SMM divided by the height squared. We used SMMI and BMI to divide the women into four groups: (1) a non-sarcopenia group (NSG): women SMMI ≥ 6.42 kg/m^2^ and BMI < 25 kg/m^2^; (2) a sarcopenia group (SG): women SMMI < 6.42 kg/m^2^ and BMI < 25 kg/m^2^; (3) a sarcopenic obesity group (SOG): women SMMI < 6.42 kg/m^2^ and BMI ≥ 25 kg/m^2^; and (4) an overweight/obesity group (NSG): women SMMI ≥ 6.42 kg/m^2^ and BMI ≥ 25 kg/m^2^.

The laboratory measurements were carried out as part of clinical routine and then abstracted from the medical records. The values of creatinine, C-reactive protein, hemoglobin, absolute lymphocyte count, total cholesterol, total proteins, serum albumin, and globulin were included [14].

We evaluated the health-related quality of life (HRQoL) using validated questionnaires, namely the Mexican-Spanish version of (A) the EORTC (European Organization for Research and Treatment of Cancer) QLQ C-30 (core questionnaire, Fayers PM, 2001) and (B) the EORTC QLQ-CX24 Cervix Uteri-Cancer-specific module [15,16].

The EORTC QLQ-C30 has six multi-item scales for patient functioning (global health status and physical, role, emotional, cognitive, and social functioning). Nine single-item scales describe symptoms (fatigue, nausea/vomiting, pain, dyspnea, insomnia, loss of appetite, constipation, diarrhea, and financial difficulties) [16].

The specific module for cervical cancer (EORTC QLQ-CX24) has four multi-item scales to evaluate functional scales (body image, sexual activity, sexual enjoyment, and sexual functionality) and five single-item symptom scales (symptom experience, lymphoedema, peripheral neuropathy, and menopausal and sexual concerns) [17,18]. For both questionnaires, each multi-item scale contains a different set of items, and no item is contained in more than one scale. All the scales are structured similarly.

We followed the instructions in the EORTC QLQ-C30 Evaluation Guide to calculate the final score. The first step was to calculate the mean of the items (that of each item contributing to the construction of the scale) and obtain the raw score. After that, we applied the linear transformation formula to standardize the raw score and transform it into a scoring system ranging from 0 to 100 [16]. A higher score on a functioning scale represents a better quality of life. In contrast, a higher score on the symptom scales indicates severe symptoms/problems and a worse quality of life.

Data were collected using the Excel package, and the statistical analysis was performed using the IBM SPSS Statistics Version 28 (IBM, Armonk, NY, USA) software package. The variables with a parametric distribution were described as means and standard deviations, and differences between groups were calculated by Student’s *t*-test or the ANOVA test. For the variables with nonparametric distributions, we calculated the median and interquartile intervals and used the Mann–Whitney U test and the Kruskal–Wallis test to identify differences between groups. The categorical variables were expressed as proportions and percentages, and the chi-square test or Fisher’s exact test was used to compare them between the patients with normal and low PA. Cronbach’s alpha value was used to assess the reliability in the case of the multi-item scales of the EORTC questionnaires. All statistical analyses were two-tailed, and a *p*-value < 0.05 was considered significant.

## 3. Results

The mean age of the 65 women with UCCa was 53 (12.43), and the mean PA was 4.48 (0.74; CI 95% = 2.50 to 4.90). The predominant histological and clinical features were squamous cancer (*n* = 40; 61.5%), followed by adenocarcinoma in 20 women (30.8%), adenosquamous in 2 women, and undetermined in 3 women. Clinical stage (CS) I accounted for 30.8% of the cases (*n* = 20), while 22 patients had CS II (33.8%), 11 women had CS III, and another 11 had CS IV (16.9%). Most women had localized advanced cancer (67.6%). Obesity was the predominant body composition phenotype (*n* = 25; 38.4%).

### 3.1. Comparisons between Normal PA and Low PA in UCCa Women

#### 3.1.1. Clinical and Anthropometric Characteristics

Twenty-nine (44.6%) women had a low PA. We found no differences in the histological and clinical features between the low-PA and normal-PA groups (Table 1).

The body composition phenotype analysis showed a higher incidence of sarcopenia (66.7%) and sarcopenic obesity (88.9%) in the low-PA group than in the normal-PA group (*p* = 0.000). In contrast, women with obesity were more common in the normal-PA group (*p* = 0.000) (Table 1).

The SMMI showed significant differences between low (5.90 ± 1.31 kg/m^2^) and normal PA (7.32 ± 1.13 kg/m^2^), but not in BMI or visceral fat (Table 2).

#### 3.1.2. Biochemical Indicators

There were significant differences between low and normal PA in the biochemical markers, including the C-reactive protein (15.84 ± 19.63 versus 6.82 ± 5.02 mg/dL, *p* = 0.022), glucose (125.4 ± 88.19 versus 88.8 ± 23.1 mg/dL, *p* = 0.021), total protein (7.28 ± 0.47 versus 7.67 ± 0.59 g/dL, *p* = 0.006), and albumin (3.9 ± 0.39 versus 4.37 ± 0.30 g/dL, *p* = 0.000).

### 3.2. HRQoL (EORTC QLQ-C30 and EORTC QLQ-CX24) of Cervix–Uteri Cancer Patients

The EORTC QLQ-C30 questionnaire showed a high reliability on the multi-item scale, with the following scores: global health status (α Cronbach = 0.842); physical (α = 0.709), role (α = 0.936), emotional (α = 0.800), cognitive α = (0.752), and social (α = 0.678) functioning scales; and fatigue (α = 0.878), nausea and vomiting (α = 0.718), and pain (α = 0.713) symptom scales.

The α Cronbach values in the QLQ-CX24 instrument were as follows: symptom experience α = 0.693, body image α = 0.852, and sexual functioning α = 0.799

#### 3.2.1. EORTC QLQ-C30

The functioning scales in the normal-PA group had median scores higher than 80, with the exception of the global health status/quality of life (75.00 (58.33–100.00)) and emotional (66.67 (41.67–91.67)) functioning scales. We found no differences in the functioning scales scores compared to the low-PA group.

Higher scores on the symptom scales were observed for fatigue, pain, loss of appetite, and financial difficulties. The low-PA group had higher scores on the loss of appetite symptom scale than the normal-PA group (*p* = 0.005).

#### 3.2.2. EORTC QLQ-CX24

The body image and sexual/vaginal functioning symptom scales showed higher scores in the EORTC QLQ-CX24 questionnaire. The scores on the menopausal symptom scale were higher in the normal-PA group than in the low-PA group (*p* = 0.039) (see Table 3).

### 3.3. Survival Status among the Cohort of Cervix–Uteri Cancer Patients

#### Survival

The survival of the cohort of women with cancer showed no significant differences according to the PA group (Table 4).

## 4. Discussion

In cancer patients, body composition analysis has been proven to be a better indicator of nutritional status than BMI. BIA provides a tool for measuring the body composition through a reproducible technique that can be performed quickly and easily at the bedside. It was used by our research team for patients with other types of cancer and has been proven to be effective among these populations [8,11].

PA, a BIA-subrogated measure, represents the resistance created by body fluids and cell membranes (capacitance) in the human body. It is correlated positively with capacitance and negatively with resistance [2], being considered an excellent marker of the cell membrane function. A low PA is associated with decreased cell integrity or cell death/apoptosis, while a high PA is associated with an intact cell and better cell membrane function [7,8,17].

The evidence presented by the cohort investigated here shows the associations of the phase angle and nutritional status with survival, the quality of life, body composition phenotype, biochemical biomarkers, and histological characteristics.

In our previous studies, we used the ROC curve tool to estimate the cutoff for PA. We used the same layout for the women who were evaluated in the present study. The normal phase angle cut-off was set to 4.45, which coincides with the Youden index of the ROC curve (cut-off point with the maximum Kolmogorov–Smirnoff), corresponding to a PA of 4.45 and an overall model quality score of 0.55 (sensitivity 0.600; specificity 0.533).

The PA values of healthy groups can vary between different racial groups (higher values in African and Caucasian groups and lower values in Asian and Latino populations), genders (higher values in men than in women), and ages (lower values after 60 years) [8,19].

Healthy people have a PA between 4 and 10 degrees, but this differs according to sex, age, weight, BMI, and fat percentage categories. These reference values can provide a basis for phase angle evaluations in the clinical setting [8,16,20,21].

The PA values of a Mexican study population were evaluated based on healthy women aged 18–82 years with a BMI between 18–31 kg/m^2^. The PA had a mean of 6.36° ± 0.97° [13,22]. Our current results yielded data consistent with our previous findings and the findings of other Mexican research groups, who found that Mexican women have a lower PA than other demographics, such as European, Afro-descended, and even Asian women [7,23].

The evaluation of PA aids in the follow-up of oncological chronic diseases, and it is a crucial tool for predicting the outcomes, frailty, functionality, adverse prognosis, response to treatment, and survival/mortality of UCCa women [19].

A secondary care hospital in San Luis Potosi, Mexico, evaluated 70 women in an observational cross-sectional study to assess the PA and body composition of women with a diagnosis of UCCa using electrical vector bioimpedance. The results showed a PA mean of 4.66 ± 0.87° (with a range of 2.9° to 6.2°). In our results, a low PA score was correlated with worse clinical stages (III and IV *n* = 20; 69%) and worse body composition phenotype (sarcopenia and sarcopenic obesity, see Table 1) [7,8].

Risk cofactors may explain the higher prevalence of obesity-associated low PA in our UCCa patients, such as smoking, malnutrition, metabolic disorders, and a sedentary lifestyle [8,11,20,24,25]. All these cofactors are associated with an impaired immune response and inflammatory mediators, which were significantly altered in our studied population (see Table 2, C-reactive protein values in women with PA < 4.45: 15.84 mg/L versus 6.82, observed in women with PA over the cutoff; *p* = 0.022).

Another mechanism identified was the percentage of adipose tissue. The number of adipose cells is related to premalignant and malignant tissue transformation in obese women. Chronic systemic inflammation due to obesity induces cell membrane damage and is also a metabolic source of inflammatory mediators (see Table 2) [8,9,17,23,26,27].

A meta-analysis published in 2016 demonstrated the link between cervical cancer and obesity. Based on the findings, obese women have a slightly increased risk of developing uterine cervical carcinoma [28]. In our current study, we identified obesity in 38.4% of the women with UCCa and in 27.7% of those with sarcopenic obesity. It is important to emphasize that the phenomenon of sarcopenic obesity is associated with a PA of <4.45 (see Table 1).

In these conditions, it is important to highlight the increased likelihood that in an ethnic or geographical group with a high prevalence of overweight–obesity (female rates in Mexico reach over 70%), sarcopenia may be associated with obesity (sarcopenic obesity), which aggravates the inflammatory conditions and causes a loss of functionality and quality of life (see Table 3) and worse clinical outcomes (see Table 2).

In UCCa, the histological type of adenocarcinoma usually responds poorly to treatment compared to squamous cell carcinoma. In our population, we observed a relationship between the body composition phenotype and histological types. Obesity was more frequently associated with adenocarcinoma and adenosquamous cervical carcinoma (14/34 (41.2%)) than squamous cell carcinoma, and sarcopenic obesity was associated with 22.2% of cases of adenocarcinoma and adenosquamous cervix–uteri carcinoma.

The biological behavior of UCCa in this group of women with obesity and sarcopenic obesity resembled the observed characteristics of endometrial cancer with diabetes and insulin resistance related to obesity.

Future studies of women with UCCa will be crucial in order to determine the interaction of the genotype of HPV infection with the overweight–obesity phenomenon, compared to the absence of HPV infection in the different histological variants of cervical cancer, and to rule out endometrial cancer with cervical invasion [20,29,30].

Carriers and survivors of UCCa show a reduction in their physical autonomy in terms of functionality due to a loss of muscle mass and the presence of symptoms related to the pathology and its treatment, and this situation directly affects their quality of life [16,18].

Based on our results, we identified the strong effect of the median value of the physical functioning scale in women with a low PA (80.0 versus 93.3 in cases of normal PA). The pattern is consistent with the symptom scale scores for fatigue (33.3 versus 22.2), pain (16.7 versus 0.00), loss of appetite (33.3 versus 0.00), and financial difficulties (66.6 versus 33.3), all with significant differences.

Based on the observed differences in the scores on the functional and symptom scales of the EORTC QLQ-C30 questionnaire among women with cervical cancer, we can hypothesize that malnutrition, coupled with cancer progression, surgery, and adjuvant therapy, together with their side effects, profoundly impairs quality of life [31].

In women with previously deteriorated HRQoL due to overweight or obesity and a loss of muscle mass, PA could be a clinical predictor of prognosis. Based on this, early nutritional counseling and support to guide physical therapy and exercise programs for overweight and obese patients could help to improve the HRQoL and functionality of patients with UCCa [31,32].

## 5. Conclusions

The main finding of the present study is the interaction between PA and obesity as critical cofactors in the UCCa adeno and adenosquamous histologic variants, to a greater extent than cervical squamous cell carcinoma.

The current results offer a basis for evaluating the interaction of the body composition with the HPV infection genotype as a risk factor for the development of cervical adeno and adenosquamous variants of UCCa in a way that resembles endometrial cancer. We conclude that this cohort is a starting point for the assessment of the consistency of obesity as a factor in cervical adenocarcinoma.

## Figures and Tables

**Table 1 healthcare-11-00246-t001:** Clinical characteristic of uterine cervical cancer patients.

Clinical Characteristic	Phase Angle < 4.45° *n* (% in a Specific Group)	Phase Angle ≥ 4.45° *n* (% in a Specific Group)	Total *n* (% Total Patients)	*p*-Value *
Histology				
Indeterminate	1 (33.3%)	2 (66.7%)	3 (4.7%)	
Squamous	18 (45.0%)	22 (55%)	40 (61.5%)	
Adenosquamous	2 (100%)	0 (0.0%)	2 (3.1%)	
Adenocarcinoma	8 (40%)	12 (60.0%)	20 (30.8%)	
Total	29 (44.6%)	36 (55.4%)	65 (100%)	0.422
Clinical Stage				
I	8 (40%)	12 (54.5%)	20 (30.8%)	
II	8 (36.4%)	14 (63.6%)	22 (33.8%)	
III	6 (54.5%)	5 (45.5%)	11 (16.9%)	
IV	6 (54.5%)	5 (45.5%)	11 (16.9%)	
Undetermined	1 (100%)	0 (0.0%)	1 (1.6%)	
Total	29 (44.6%)	36 (55.4%)	65 (100%)	0.575
Phenotype by Body Composition				
No sarcopenia	0 (0.0%)	4 (100%)	4 (6.2%)	
Sarcopenia	12 (66.7%)	6 (33.3%)	18 (27.7%)	
Sarcopenic obesity	17 (94.4%)	1 (5.6%)	18 (27.7%)	
Obesity	0 (0%)	25 (100%)	25 (38.4%)	
Total	29 (44.6%)	36 (55.4%)	65 (100%)	0.000

* Significant *p*-value < 0.05. Chi-square test.

**Table 2 healthcare-11-00246-t002:** Comparison of the anthropometric and biochemical parameters between uterine cervical cancer patients with low PA and normal PA.

Anthropometrical and Biochemical Indicators	Phase Angle < 4.45° *n* = 29	Phase Angle ≥ 4.45° *n* = 36	*p*-Value
Age and Anthropometrical Indicators			
Age	59 (12.0)	48 (11.0)	0.000 *
Phase angle	3.84 (0.54)	4.99 (0.39)	0.000
Body Mass Index (BMI)	27.34 (6.36)	28.30 (5.63)	0.526
Visceral Fat	1.8 (0.8)	1.9 (0.6)	0.382
Lean Mass Percentage	57.2 (8.70)	60.10 (7.20)	0.147
Skeletal Muscle Mass Index (SMMI)	5.90 (1.31)	7.32 (1.13)	0.000 *
Biochemical Indicators			
Hemoglobin (g/dL)	11.80 (2.1)	12.3 (1.5)	0.237
Erythrocyte Sedimentation Rate (mm/seg)	31.27 (7.7)	29.08 (10.4)	0.339
C-Reactive Protein (mg/L)	15.84 (19.63)	6.82 (5.02)	0.022 *
Creatinine (mmol/L)	72.66 (26.19)	69.87 (16.53)	0.607
Glucose (mg/dL)	125.39 (88.19)	88.78 (23.08)	0.021 *
Total Proteins (g/dL)	7.28 (0.47)	7.67 (0.59)	0.006 *
Albumin (g/dL)	3.9 (0.39)	4.37 (0.30)	0.000 *
HDL-Cholesterol (mg/dL)	50.55 (15.95)	51.74 (15.57)	0.770
Triglycerides (mg/dL)	158.14 (65.10)	235.48 (234.38)	0.102
Total cholesterol (md/dL)	194.86 (50.91)	208.49 (36.29)	0.223

* Significant *p*-value < 0.05. Student’s *t* test.

**Table 3 healthcare-11-00246-t003:** Comparison of EORTC QLQ-C30 and EORTC QLQ CX 24 scores between uterine cervical cancer patients with low PA and normal PA.

Scores for the QLQ Scales	Phase Angle < 4.45° *n* = 29 Median (P25-P75)	Phase Angle ≥ 4.45° *n* = 36 Median (P25-P75)	*p*-Value
EORTC QLQ-C30 (SCORE 0–100)			
Global Health Status/Quality of Life	83.33 (75.00–100.00)	75.00 (58.33–100.00)	0.289
Physical Functioning	80.00 (66.67–93.33)	93.33 (73.33–100.00)	0.102
Role Functioning	100.00 (33.33–100.00)	100.00 (100.00–100.00)	0.057 *
Emotional Functioning	83.33 (58.33–100.00)	66.67 (41.67–91.67)	0.231
Cognitive Functioning	100.00 (66.67–100.00)	100.00 (66.67–100.00)	0.716
Social Functioning	100.00 (66.67–100.00)	100.00 (83.33–100.00)	0.548
Fatigue	33.33 (0.0–66.67)	22.22 (0.0–33.33)	0.203
Nausea and Vomiting	0.0 (0.0–16.67)	0.0 (0.0–0.0)	0.263
Pain	16.67 (0.0–50.00)	0.0 (0.0–33.33)	0.358
Dyspnea	0.0 (0–33.33)	0.0 (0.0–33.33)	0.980
Insomnia	0.0 (0.0–66.67)	0.0 (0.0–66.67)	0.959
Loss of Appetite	33.33 (0.0–100.00)	0.0 (0.0–0.0)	0.005 *
Constipation	0.0 (0–33.33)	0.0 (0.0–0.0)	0.275
Diarrhea	0.0 (0.0–66.67)	0.0 (0.0–0.0)	0.145
Financial Difficulties	66.67 (0.0–66.67)	33.33 (0.0–100.00)	0.725
EORTC QLQ CX 24 (SCORE 0–100)			
Body Image	100.00 (77.80–100.00)	100.00 (66.70–100.00)	0.830
Sexual Activity	0.0 (0.0–0.0)	0.0 (0.0–33.33)	0.200
Sexual Enjoyment	27.70 (27.70–27.70)	27.70 (27.70–27.70)	0.191
Sexual/Vaginal Functioning	62.20 (62.20–62.20)	62.20 (62.20–62.20)	0.404
Symptom Experience	12.10 (6.10–18.20)	9.10 (3.0–24.20)	0.444
Lymphoedema	0.0 (0.0–33.33)	0.0 (0.0–33.33)	0.645
Peripheral Neuropathy	0.0 (0.0–33.33)	0.0 (0.0–33.33)	0.647
Menopausal Symptoms	0.0 (0.0–33.33)	0.0 (0.0–100.0)	0.039 *
Sexual Concern	0.0 (0.0–66.70)	0.0 (0.0–100.0)	0.572

* Significant *p*-value < 0.05. Mann–Whitney *U* test. Nonparametric distribution values. Median (interquartile interval).

**Table 4 healthcare-11-00246-t004:** Survival, death, and loss of follow-up.

Survival Status				
Clinical Characteristic	Phase Angle < 4.45° *n* (% in a Specific Group)	Phase Angle ≥ 4.45° *n* (% in a Specific Group)	Total *n* (% Total Patients)	*p*-Value *
Alive	18 (37.8%)	28 (62.2%)	46 (100.0%)	
Death	10 (66.7%)	5 (33.3%)	15 (100.0%)	
Lost to follow-up	2 (50.0%)	2 (50.0%)	4 (100.0%)	
Total	29 (44.6%)	36 (55.4%)	65 (100.0%)	0.148

* Significant *p*-value < 0.05. Chi-square *t*-test.

## Data Availability

The datasets generated and analyzed in the current study are not publicly available because they are the property of the Instituto Mexicano del Seguro Social. Institutional and federal bodies restrict unlimited access to personal data, but they are available from the corresponding authors on reasonable request with prior authorization from the institution.

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
