# Peer review of "Phase Angle and Nutritional Status: The Impact on Survival and Health-Related Quality of Life in Locally Advanced Uterine Cervical Cancer"

_healthcare, 2023, doi:10.3390/healthcare11020246_

Round 1

Reviewer 1 Report

The conclusion drawn in the abstract seems to not have any association with the main objective of this study. Please revise.Your whole work is about 65 patients. We understand that you have a main cohort of 136 cases but its description does not add any value to the article.

Why is their only 64 patients in table 2 and not 65 ?

In your table p value = 0.000 is not informative please indicate you result in  a different way.

Author Response

Cover Letter

January 03, 2022

Prof. Dr. George Moschonis

Guest Editor

School of Allied Health, Human Services and Sport, La Trobe University, Melbourne 3086, Australia

Interests: nutritional epidemiology; nutritional assessment; nutritional counselling; functional foods

Dr. Anj Reddy E-Mail Website

Guest Editor

1)Mary MacKillop Institute for Health Research, Australian Catholic University, Melbourne 3000,

Australia; 2) School of Allied Health, Human Services and Sport, La Trobe University, Melbourne 3086,

Australia Interests: dietary interventions; anti-inflammatory diets; inflammation; non-alcoholic fatty

liver disease; metabolic disorders; nutrigenomics

Reviewers Comments to: Cervix uteri cancer locally advanced: Nutritional State, Phase Angle and

biochemical markers impact on survival and Health related Quality of Life

Comments based on the Review Report Form Reviewer 1

Open Review

English language and style

( ) English very difficult to understand/incomprehensible

(x) Extensive editing of English language and style required

  • (  ) Moderate English changes required

  • (  ) English language and style are fine/minor spell check required

  • (  ) I don't feel qualified to judge about the English language and style

    Yes

Can be improved

Must be Not improved applicable

() ()

() () (x) () () () () () () ()

Does the introduction provide sufficient background and include all relevant references?
Are all the cited references relevant to the research?

Are the results clearly presented?
Are the conclusions supported by the results?

(x) ()

(x) ()

  • ()  ()

  • ()  (x)

(x) () (x) ()

Is the research design appropriate?

Are the methods adequately described?

In response to the request to realize an extensive editing of English language and style required, we decided sending the manuscript to the editorial services provided by English Editing Department, from MDPI Editorial.

We decide change the title to enhance the English editing and for a better understanding of the paper aim:

We enhance the research design and the methods considering the extended comments that Reviewer 1 express below in her/his observations.

“Phase angle and nutritional status: the impact on survival and health-related quality of life in locally

advanced uterine cervical cancer“

Comments and Suggestions for Authors (highlighted in yellow) and response from Authors (marked with underlined phrases or paragraphs)
The conclusion drawn in the abstract seems to not have any association with the main objective of this study. Please revise. Your whole work is about 65 patients.

but its description does not add any value to the article.

We understand that you have a main cohort of 136 cases

Response: We have eliminated the description of the cohort with whole the 136 cases, and preserved the data

from the 65 studied cases.

Why is their only 64 patients in table 2 and not 65?

In your table p value = 0.000 is not informative please

indicate you result in a different way.

Response: We checked and corrected Table 2. In our database we have 36 women with a phase angle above

4.45°, and in a typo we wrote 35 instead of 36, these are the analyzed patients. Above the p-value = 0.000, the

p-value means that the differences between the groups are highly significant. In this way, and after discussion

of the issue among corresponding authors, middle author, and co-authors, we decide to keep the p-value as it

is in the tables

We believe that this article is relevant to put in context all the above mentioned factors in Countries with a high prevalence of overweight and obesity publishing scope and will be of interest to its readership. This manuscript has not been published elsewhere and is not under consideration by another journal. We have approved the manuscript and agree with submission to the special. The study was supported by Instituto Mexicano del Seguro Social and the Centro Universitario de Ciencias de la Salud – Universidad de Guadalajara, using their own resources. There are no conflicts of interest to declare.

We look forward to hearing from the editorial board at their earliest convenience. Sincerely,

Dr. Brenda-Eugenia Martínez-Herrera PhD. Hospital “Dr. Francisco Padrón Puyou”, Órgano de Operación Administrativa Desconcentrada San Luis Potosi, IMSS, San Luis Potosi 78250, Mexico
e-mail: [email protected],

Dr. Daniel Sat-Muñoz MD, MBA

950 Sierra Mojada, Gate 7, Building C, 1st level, Colonia Independencia, Guadalajara, Jalisco 44340, México
+52 33 3668 3000 extensión 31611
+52 33 1349 6920 (cell phone) [email protected], alternative email: [email protected]

Reviewer 2 Report

González-Barba et al. report an interesting assessment of obesity, nutritional markers, and cervical cancer. Authors state that phase obesity is more common in adeno- types of cervical cancer than in squamous cell carcinoma. This report has potential, but currently the results are buried and need to be made more clear. The introduction should introduce ideas needed for understanding and report what is found in the literature without subjective opinions. After reading the introduction it is unclear where the paper is going, if all of the results are going to be compared to the phase angle, this should be the focus of the title, abstract, and introduction. However if this is not the focus of the paper, the results sections need to be recalculated to focus on what is clinically and statistically relevant.

Comment 1: Low and high phase angle measurements should have an explanation in the abstract. It can be brief, but some introduction to the idea of cancer related sarcopenic obesity or malnourishment would be helpful to understand the key points of the paper in the abstract. The only description of phase angle is in the discussion, this should come earlier to introduce the reader to the idea.

Comment 2: there is a typo on line 129 "kwon" should be "known"

Comment 3: The introduction makes some good points but should be refined to be less subjective. When the authors mention "bibliographic search" or "bibliographic evidence" are they referring to published literature review? If the authors did a formal literature review they should be more clear about it, or if they didn't they should leave out speculation about the state of current literature.

Comment 3: The authors state that these are the results of the first 65 patients. Is this how many patients were left after the exclusion criteria? Are there plans to collect the body composition data on the rest of the cohort? If so this should be included here. 

Comment 4: The methods should explain what phase angle is in detail since all of the results are using this metric.

Comment 5: The result tables are all reported relative to the phase angle measurements, however it is not clear why phase angle was used. Is the purpose of the study to validate the use of this metric? A more appropriate comparison may be to a clinical outcome or QOL measure. Consider making the explanation of the tables more clear.

Comment 6: Line 348 "metabolic disturbs" seems like a typo, I think they are referring to metabolic syndrome.

Comment 7: Line 386 should say "HPV" not "HVP"

Comment 8: Line 398, unclear what is meant by "we found a profound affection in the median value".

Comment 9: In the conclusion authors state "interaction with the HPV infection as a risk factor to develop cervical adenocarcinoma such as endometrial cancer". Is endometrial cancer associated with HPV infection? Authors should double check and cite sources. 

Comment 10: The methods should detail what statistical tests were used and why.

Author Response

Cover Letter

January 03, 2022

Prof. Dr. George Moschonis

Guest Editor

School of Allied Health, Human Services and Sport, La Trobe University, Melbourne 3086, Australia

Interests: nutritional epidemiology; nutritional assessment; nutritional counselling; functional foods

Dr. Anj Reddy E-Mail Website

Guest Editor

1)Mary MacKillop Institute for Health Research, Australian Catholic University, Melbourne 3000,

Australia; 2) School of Allied Health, Human Services and Sport, La Trobe University, Melbourne 3086,

Australia Interests: dietary interventions; anti-inflammatory diets; inflammation; non-alcoholic fatty

liver disease; metabolic disorders; nutrigenomics

Reviewers Comments to: Cervix uteri cancer locally advanced: Nutritional State, Phase Angle and

biochemical markers impact on survival and Health related Quality of Life

Comments based on the Review Report Form Reviewer 2

Open Review

English language and style

( ) English very difficult to understand/incomprehensible
( ) Extensive editing of English language and style required
(x) Moderate English changes required
( ) English language and style are fine/minor spell check required
( ) I don't feel qualified to judge about the English language and style

Yes

Can be improved

Must be Not improved applicable

(x) ()

() () () () () () (x) () () ()

Does the introduction provide sufficient background and

() ()

(x) ()

  • ()  (x)

  • ()  (x)

  • ()  ()

  • ()  (x)

include all relevant references?

Are all the cited references relevant to the research?

Is the research design appropriate?

Are the methods adequately described?

Are the results clearly presented?

Are the conclusions supported by the results?

Comments and Suggestions for Authors

González-Barba et al. report an interesting assessment of obesity, nutritional markers, and cervical cancer. Authors state that phase obesity is more common in adeno- types of cervical cancer than in squamous cell carcinoma. This report has potential, but currently the results are buried and need to be made more clear. The

introduction should introduce ideas needed for understanding and report what is found in the literature without

subjective opinions

. After reading the introduction it is unclear where the paper is going, if

However if this is not the focus of the paper, clinically and statistically relevant.

all of the results are

going to be compared to the phase angle, this should be the focus of the title, abstract, and introduction

.

the results sections need to be recalculated to focus on what is

We realize an extensive editing of English language and style required, we decided sending the manuscript to

the editorial services provided by English Editing Department, from MDPI Editorial.

We decide too, change the title to enhance the English editing and for a better understanding of the paper aim:

Comment 1: . It can be would be

helpful to . The only description of phase angle is in the discussion, .

“Phase angle and nutritional status: the impact on survival and health-related quality of life in locally

advanced uterine cervical cancer“

Low and high phase angle measurements should have an explanation in the abstract

brief, but some introduction to the idea of cancer related sarcopenic obesity or malnourishment

understand the key points of the paper in the abstract

this should come earlier to introduce the reader to the idea

Response: We have highlighted the importance of the phase angle in the summary and introduction in

conjunction with the current results.

Comment 2: there is a typo on line 129 "kwon" should be "known"

Comment 3: The introduction makes some good points but authors

should leave out speculation about the state of current literature.

. When the , or if they didn't they

We realize an extensive editing of English language and style required, we decided sending the manuscript to

the editorial services provided by English Editing Department, from MDPI Editorial.

should be refined to be less subjective

mention "bibliographic search" or "bibliographic evidence" are they referring to published literature

review? If the authors did a formal literature review

they should be more clear about it

Response: We rewrote the introduction section to clarify the concepts before submitting them to the English

edition

Comment 3: The authors state that these are the results of the first 65 patients. Is this how many patients were left after the exclusion criteria? Are there plans to collect the body composition data on the rest of the cohort? If

.

Comment 4: The methods should explain what phase angle is in detail since all of the results are using this metric.

Response: We rewrote the methods section to clarify the methodology before submitting them to the English

edition

Comment 5: The result tables are all reported relative to the phase angle measurements, however it is not clear why phase angle was used. Is the purpose of the study to validate the use of this metric? A more appropriate comparison may be to a clinical outcome or QOL measure. Consider making the explanation of the tables more clear.

Comment 6: Line 348 "metabolic disturbs" seems like a typo, I think they are referring to metabolic syndrome.

Comment 7: Line 386 should say "HPV" not "HVP"

so this should be included here

Response: We have eliminated the description of the cohort with whole the 136 cases, and preserved the data

from the 65 studied cases.

Response: We checked the tables and statistical methods and we explained it more clearly, we corrected

some spot errors that we found during the editing process

Response: We realize an extensive editing of English language and style required, we decided sending the

manuscript to the editorial services provided by English Editing Department, from MDPI Editorial.

Response: We realize an extensive editing of English language and style required, we decided sending the

manuscript to the editorial services provided by English Editing Department, from MDPI Editorial. The typo

mistakes were corrected.

Comment 8: Line 398, unclear what is meant by "we found a profound affection in the median value".

Comment 9:

.

Comment 10: The methods should detail what statistical tests were used and why.

We believe that this article is relevant to put in context all the above mentioned factors in Countries with a high prevalence of overweight and obesity publishing scope and will be of interest to its readership. This manuscript has not been published elsewhere and is not under consideration by another journal. We have approved the manuscript and agree with submission to the special. The study was supported by Instituto Mexicano del Seguro Social and the Centro Universitario de Ciencias de la Salud – Universidad de Guadalajara, using their own resources. There are no conflicts of interest to declare.

We look forward to hearing from the editorial board at their earliest convenience. Sincerely,

Response: We realize an extensive editing of English language and style required, we decided sending the

manuscript to the editorial services provided by English Editing Department, from MDPI Editorial. The

redaction mistakes were corrected.

In the conclusion authors state "interaction with the HPV infection as a risk factor to develop

cervical adenocarcinoma such as endometrial cancer". Is endometrial cancer associated with HPV infection?

Authors should double check and cite sources

Response: In the conclusion section, we clarified that we propose to assess low PA and obesity as cofactors

associated with specific HPV genotypes in the development of adeno and adenosquamous

Response: Answer: We have corrected and clarified the statistical test errors, in the footnote of the tables and

in the methodology section too.

Round 2

Reviewer 1 Report

No comment